# Natural Foraging Selection and Gut Microecology of Two Subterranean Rodents from the Eurasian Steppe in China

**DOI:** 10.3390/ani14162334

**Published:** 2024-08-13

**Authors:** Zhenghaoni Shang, Kai Chen, Tingting Han, Fan Bu, Shanshan Sun, Na Zhu, Duhu Man, Ke Yang, Shuai Yuan, Heping Fu

**Affiliations:** 1College of Grassland Resources and Environment, Inner Mongolia Agricultural University, Hohhot 010011, China; shangzhn1997@163.com (Z.S.); ckai1414@163.com (K.C.); hantingting1015@163.com (T.H.); bufanimau@163.com (F.B.); sunshanshan557@163.com (S.S.); jvn0397@163.com (N.Z.); 2Key Laboratory of Grassland Rodent Ecology and Pest Controlled, Hohhot 010011, China; 3Key Laboratory of Grassland Resources, Ministry of Education, Hohhot 010011, China; 4College of Agriculture, Hulunbuir University, Hulunbuir 021000, China; mantou08@126.com; 5Alxa League Meteorological Bureau, Alxa 750300, China; yangke19961006@163.com

**Keywords:** *Myospalax*, gut microbiota, diet, cellulose degradation, microbial niche

## Abstract

**Simple Summary:**

Rodents, as the most abundant group of mammals, offer a unique opportunity to study the relationship between diet and host gut microecology. In this study, we focused on two species of the genus *Myospalax* in the Eurasian steppes in China: *M. psilurus* and *M. aspalax*. These specialized herbivorous rodents have adapted to a subterranean lifestyle and exhibit distinct dietary choices. Our study aimed to investigate how their diet composition is related to their gut microbial communities and whether there are unique indicator species for their gut microecology. Using 16S amplicon technology and macro-barcoding technology, we analyzed the gut bacterial communities and diet composition of the two zokor species. We found that *M. psilurus* had a higher diversity of gut microbial bacterial communities compared to *M. aspalax*. The two species also possessed different gut bacterial indicator species, with unique relationships between their diet and gut microbes. Our results provide new insights into the adaptation of zokors to long-term subterranean life and shed light on their cellulose degradation abilities and ecological niches.

**Abstract:**

As the most abundant group of mammals, rodents possess a very rich ecotype, which makes them ideal for studying the relationship between diet and host gut microecology. Zokors are specialized herbivorous rodents adapted to living underground. Unlike more generalized herbivorous rodents, they feed on the underground parts of grassland plants. There are two species of the genus *Myospalax* in the Eurasian steppes in China: one is *Myospalax psilurus*, which inhabits meadow grasslands and forest edge areas, and the other is *M. aspalax*, which inhabits typical grassland areas. How are the dietary choices of the two species adapted to long-term subterranean life, and what is the relationship of this diet with gut microbes? Are there unique indicator genera for their gut microbial communities? Relevant factors, such as the ability of both species to degrade cellulose, are not yet clear. In this study, we analyzed the gut bacterial communities and diet compositions of two species of zokors using 16S amplicon technology combined with macro-barcoding technology. We found that the diversity of gut microbial bacterial communities in *M. psilurus* was significantly higher than that in *M. aspalax,* and that the two species of zokors possessed different gut bacterial indicator genera. Differences in the feeding habits of the two species of zokors stem from food composition rather than diversity. Based on the results of Mantel analyses, the gut bacterial community of *M. aspalax* showed a significant positive correlation with the creeping-rooted type food, and there was a complementary relationship between the axis root-type-food- and the rhizome-type-food-dominated (containing bulb types and tuberous root types) food groups. Functional prediction based on KEGG found that *M. psilurus* possessed a stronger degradation ability in the same cellulose degradation pathway. Neutral modeling results show that the gut flora of the *M. psilurus* has a wider ecological niche compared to that of the *M. aspalax*. This provides a new perspective for understanding how rodents living underground in grassland areas respond to changes in food conditions.

## 1. Introduction

Microbes, as resident populations that colonize the mammalian body (especially the gut) [1], far outnumber mammalian somatic cells, and unique genes encoded by microbes outnumber the host’s genome 100-fold [2]. Most microorganisms residing in the gut have profound effects on host physiology and nutrition, which are critical for host health [3,4,5,6]. Thanks to the rise of human intestinal microbiome research, the emerging field of bacterial-dominated gut microecology attempts to answer a growing number of biological questions [7,8]. Recently, more and more research is shifting its focus from the gut microbiomes of humans to those of other mammals in their natural environments in order to more deeply explore the mutual adaptations and selection of the host and gut bacteria [9,10,11].

Mutually beneficial symbiosis between a host and gut microbiota is thought to have arisen through a long period of co-evolution [12]. As an important component of the host’s metabolism, intestinal microbiota can provide substrates, enzymes, and energy to the host [13]. The composition of the gut microbiome changes with the host’s physiological state, food, and habitat [14]. In addition to vertical inheritance caused by genetic factors, diet is a major determinant of gut microbiota composition [15,16,17]. Rodents, as the most abundant group of mammals, have a very rich ecotype [18,19], making them ideal for studying the relationship between diet and host gut microecology [20]. 

Food habits are an important expression of animals’ adaptation to their environment, and profoundly affect their role and function in the ecosystem [21,22]. Animal food habits not only determine their access to energy and nutrients [23], but also reflect the symbiotic relationships between animals and other organisms [24,25]. In addition, animal food habits can be an indicator of ecosystem health [26,27]. The presence of a large number of species in the food composition of herbivorous animals may be related to the structure and energy flow of the plant community in the habitat [28,29], and the food selection pattern can reflect the phenology and availability of the plant community [30]. The “diets” of animals often include many contents, such as food composition [31,32], food nutrients [33,34], and feeding patterns [35,36]. In this study, we focused on the dietary choices of two species of zokors in their natural environment. Therefore, the term “diet” is defined in this paper as the species that the rodents feed on and their types and proportions.

As r-strategists, rodents have evolved diverse food habits to adapt to various complex environments [37,38]. Some of these have evolved to be highly specialized, such as animals in the subfamily Myospalacinae (zokors). Unlike generalized herbivorous rodents, zokors are specialized herbivorous rodents adapted to living underground, which feed on the underground parts of grassland plants [39]. The subfamily *Myospalax* contains two genera, *Myospalax* and *Eospalax* [40]. There are two species of the genus *Myospalax* in China: one is the North China Zokor (*M. psilurus*), which inhabits meadow grasslands and forest edge areas, and the other is the Steppe Zokor (*M. aspalax*), which inhabits typical grassland areas [41,42]. The two species diverged from their ancestors about 2.9 million years ago, and the two populations came into secondary contact about 1.7 million years ago, resulting in weak gene flow. Although *M. aspalax* experienced population shrinkage and *M. aspalax* experienced population expansion, the effective population size of *M. aspalax* is still larger than that of *M. psilurus* [43]. So far, research on these two species of zokors has mainly focused on morphology, genetics and other topics [44,45], and research on their microecology, especially research on the relationship between the dietary choices of two species and their intestinal microorganisms, is relatively rare.

Based on the ecological differences described above, we hope to compare the gut microbial profiles of these two closely related species based on dietary composition, and to describe the types of gut microbial communities of the two zokors through indicator genera. *M. psilurus* lives at the edges between forest margins and grasslands [46], where the vegetation types are more diverse, and there are more species of plants available for consumption than for *M. aspalax*. Based on the amount of plant diversity in the two zokor habitats and the expansion shown by the *M. psilurus* population, we hypothesized that *M. psilurus* has higher dietary diversity and microbial diversity than *M. aspalax*. When studying the structure and dynamics of a given community, α-diversity contributes to the understanding of species composition and interactions within the community, and it reflects the diversity and abundance of species in the sample [47]. We used α-diversity to respond to the dietary and microbial diversity of zokors. Specific evaluation indexes included ACE index, Shannon index and Pd index. Among them, ACE index was used to measure species richness, i.e., the number of species, while Shannon index and Phylogenetic diversity were used to measure species diversity [48]. As specialized herbivorous rodents, their diets are rich in fiber, which is a potential carbon source [49], and the function of the gut microbiota may be enriched in cellulose degradation. Food digestion and absorption are key processes in animals’ adaptive evolution [50]. Population expansion is inseparable from the efficient use of food. We predict that the gut flora of *M. psilurus* will be slightly better at degrading cellulose. We used amplicons to analyze the dietary and gut microbiological characteristics of two species of zokors. we aimed to: (i) clarify the food composition and gut microbiological characteristics of different hosts, (ii) clarify the relationship between food type and gut bacterial community and find indicator genera for the gut bacterial characteristics of this species, and (iii) determine the mechanism of construction of the gut bacterial community of the two species of zokors.

## 2. Materials and Methods

To address the above questions, we conducted a preliminary investigation of the intestinal bacterial communities and dietary compositions of two species of zokor, using the 16S amplicon technique in combination with the macro-barcoding technique.

### 2.1. Sample Collection

Two species of zokors were captured using the circular tongs trap method from August to September 2021 (Autumn) in the Hulunbeier meadow grassland (Chenbalhu Banner) and the typical grassland of Xilingol (Zhengxiangbai Banner) in Inner Mongolia, China. A total of 33 adult individuals in the non-breeding phase were collected, including 18 *M. psilurus* (8 males and 10 females) and 15 *M. aspalax* (8 males and 7 females) (Figure 1). The body weight of *M. psilurus* was 274.62 ± 41.92 g, and that of *M. aspalax* was 272.01 ± 69.79 g. Euthanasia was carried out by carbon dioxide inhalation in zokors. Samples of cecum contents and stomach contents were collected and then immediately stored in liquid nitrogen. The samples were transported to the laboratory on dry ice and stored at −80 °C.

### 2.2. DNA Extraction and Sequencing of Cecum and Stomach Contents

Microbiota from cecum contents and plant DNA from stomach contents were extracted using the TruSeq^TM^ DNA Sample Prep Kit (Illumina Inc., San Diego, CA, USA) according to the instructions. DNA quality was checked using a NanoDrop2000 ultra-trace spectrophotometer (Thermo Scientific, Waltham, MA, USA) and 1% agarose gel electrophoresis. The 16SrRNA V3-V4 region of the microbiota was amplified using universal primers (338F and 806R) [31], and plant root communities were assessed by amplifying trnL operons (trnL-F: 5′-CGAAATYGGTAGACGCTACG-3′ and trnL-R: 5′-CCDTYGAGTCTCTGCACCTATC-3′) [51]. Each sample was replicated three times, and the PCR products from the same sample were mixed and detected by 2% agarose gel electrophoresis. The PCR products were recovered by cutting the gel using the Axy Prep DNA Gel Recovery Kit (AXYGEN Inc., Union City, CA, USA), eluted with Tris_HCl, and detected by 2% agarose electrophoresis. Referring to the preliminary quantitative results of electrophoresis, the PCR products were detected and quantified by QuantiFluor™-ST Blue Fluorescence Quantification System (Promega Corporation, Madison, WI, USA), after which the PCR products were mixed according to the corresponding proportion of each sample according to the sequencing amount required. After constructing the clone library using the TruSeq^TM^ DNA kit (Illumina Inc., San Diego, CA, USA), high-throughput sequencing (250 bp, double-end sequencing) was performed using the Illumina Nova Seq platform.

### 2.3. Bioinformatics Analysis

The raw sequences were quality-controlled using fastp software (version 0.19.6) [52] and FLASH (version 1.2.7) software for splicing [53]. Bases with quality values below 20 in the tails of the reads were filtered, and a window of 50 bp was set. If the average quality value within the window was below 20, then the back-end bases were truncated, starting from the window, while reads below 50 bp after quality control were filtered, and reads containing N bases were removed. Based on the overlap relationship between PEreads, pairs of reads were spliced (merged) into one sequence with a minimum overlap length of 10 bp; the maximum mismatch ratio allowed in the overlap region of the spliced sequences was 0.2, and non-compliant sequences were screened out. The samples were distinguished according to the barcode and primers at the first and last ends of the sequences, and the sequences were adjusted based on the direction. The barcode needs to be an exact match, and only 2-base mismatches were allowed for the primers.

Using UPARSE software (version 7.1), OTU clustering was performed on non-repetitive sequences (excluding single sequences) according to 97% similarity, and chimeras were removed during the clustering process to obtain OTU representative sequences [54,55]. All sample sequences were levelled by sequence pumping at the minimum number of sample sequences, and the average sequence coverage (Good’s coverage) for each sample could still reach 99.09%. The RDPclassifier (version 2.11) was used to compare to the Silva16SrRNA gene database (v138) for the taxonomic annotation of intestinal microbial OTU species [56], with a confidence threshold of 70%, and to count the community composition of each sample at several different taxonomic levels. 16S functional prediction analyses were performed using PICRUSt2 (version 2.2.0) software [57]. The ny_v20221012 species classification database and the RDP Classifier2.13 software were used to classify and annotate gastric capacitance DNA macro-barcode sequences. The Species 2000 Chinese Nodal Plant Group Database [58] was used to search for plant species that might be present in the habitats of both species of zokors using the “distribution area + classification system” filters. The distribution range of plants that could potentially be consumed by the North China zokor was limited to “Inner Mongolia Autonomous Region” and “Heilongjiang Province”. The distribution of plants that could potentially be eaten by the steppe zokor was limited to “Inner Mongolia Autonomous Region” and “Hebei Province”, and the search condition was “distribution in the above areas”. The plant classification order was specified from Plantae to Magnoliopsida for screening and downloading information on plant species found in the two study habitats. The results of the downloads were cross-referenced with the macro-barcode annotations to match species with the same name and exclude species that are not reported in the habitat.

Several alpha diversity indices (ACE, Shannon, and Pd indices) were calculated using the mothur software (version 1.30.2) [48], and the Wilcoxon rank-sum test was used for analyses of between-group differences in alpha diversity. Using the data table in the tax_summary_a folder, R statistical software (version 3.3.1) was used to produce microbial community bar charts and plant community pie charts. The Wilcoxon rank-sum test was performed using R statistical software and the scipy package of python to analyze differences between plant species in the samples of the two zokor species’ stomach contents. The *p*-values were corrected using the Fdr method, and the confidence interval was calculated using the Welch’s *t*-test with a confidence level of 0.95. All plants were classified into 8 types by plant root type, and the gut bacterial communities of both zokor species were analyzed by food root type with a Mantel analysis using the vegan package (vsesion 2.4.3). IndVal was calculated using the lndval function of the labdsv package [59] to find the indicator genera of the gut bacterial communities of the two zokors, and OTU clustering information was compared with the sequenced microbial genome databases using the PICRUSt2 software to obtain the corresponding species in the KEGG database https://www.kegg.jp/ (accessed on 1 August 2023) for the functional type and abundance of the corresponding species. Differences in KEGG pathways between the two zokor species were analyzed using Statistical Analysis of Macrogenome Mapping (STAMP, vsesion 2.1.3), and the false discovery rate was controlled using the Benjamini–Hochberg procedure [60].

We used the Neutral Community Model (NCM) to assess the potential impacts of stochastic processes on the assembly of gut microbial communities in two species of zokors [61,62]. The model used here is based on neutral theory [63,64], where N denotes the metacommunity size, i.e., the total abundance of all OUTs in each sample; m denotes the community-level mobility, where a smaller value indicates that species dispersal is restricted throughout the community, and vice versa; Nm quantifies the estimate of species dispersal between communities and determines the correlation between the frequency of occurrences and the relative abundance in the area. The parameter R^2^ indicates the overall fit to the neutral model, with a higher R^2^ indicating a closer approximation to the neutral model, i.e., community construction is more influenced by stochastic processes and less by deterministic processes. The calculation of 95% confidence intervals for all fit statistics was performed by bootstrapping using 1000 Bootstrap replications. In this study, we used datasets from two species of zokors. Subsequently, the OTUs of each dataset were categorized into three partitions, depending on whether they occurred more frequently (higher partition), less frequently (lower partition), or within the 95% confidence interval predicted by the NCM (neutral partition). To analyze deviations from NCM predictions, we compared the composition, diversity, and calculated estimated mobility rate (m) of neutral and non-neutral (upper and lower) partitions of the gut bacterial community. All calculations were performed in R (version 3.2.3) using the Hmisc (version 5.0.1), minpack.lm (version 1.2.3) and getopt (version 1.20.3) packages.

## 3. Results

Pan and Core species analyses based on macro-barcoding and 16S amplicon sequencing results show that the number of Pan OTUs tends to zero as the number of samples increases, and the number of Core OTUs tends to level off as the number of samples increases (Appendix A). Such results suggest that the measured sample size is sufficient to assess the total species richness and core species number in the environment, indicating that the sample size used in this study is sufficient to reflect the objective facts.

### 3.1. Analysis of Food Composition and Dietary Differences

Based on the results of macro-barcoding sequencing, after excluding species that could not be classified and those with ultra-low abundance, there were a total of 55 food species for *M. psilurus* and *M. aspalax*. Of these, 25 species were unique to *M. psilurus*, and 4 species were unique to *M. aspalax* (Appendix A). From the macro-barcoding results, we determined the top ten species in the diets of each species of zokor. *Sanguisorba officinalis* was the main food species of *M. psilurus*, accounting for about 68% (Figure 2A). *Allium tuberosum* and *Phlomis* sp. were the main sources of food for *M. aspalax*, accounting for 38.1% and 14%, respectively (Figure 2B). The diversity of diet of *M. psilurus* was not significantly different from that of *M. aspalax*. (ACE index, *p* = 0.91; Shannon index, *p* = 0.91). The analysis of species differences in food habits shows that food components were significantly different between the two species of zokors (Figure 2C). *S. officinalis*, *A. tuberosum*, and *Thalictrum minus* were detected in the diets of both species of zokors, but at different abundances. *S. officinalis* was more abundant in the diet of *M. psilurus* than in that of *M. aspalax*, whereas *A. tuberosum* and *T. minus* were more commonly eaten by *M. psilurus* (Figure 2C). *Pimpinella* sp., *Potentilla* sp., *Viola* sp., *Bupleurum scorzonerifolium*, *Adenophora* sp., *Plantago* sp., and *Galium dahuricum* were exclusively found in the samples from *M. psilurus* (Figure 2C).

### 3.2. Sequencing Information and Gut Microbial α-Diversity

We obtained a total of 933,735 optimized sequences from 33 samples using 16S amplicon sequencing, resulting in 28,295 valid sequences with an average sequence length of 413. As the number of sequenced samples increased, the dilution curve flattened out, and the coverage rate of all samples was over 99.90%. This indicates that the amount of sequenced sequences was reasonable, and the depth of sequencing was sufficient to cover all samples (Appendix A). The 33 samples from the two species of zokors contained a total of 1280 OTUs, and the number of shared OTUs accounted for 69.14% (Figure 3A), namely, norank_f_Muribaculaceae (37.62%), Lachnospiraceae_NK4A136_group (10.43%) and unclassified_f_Lachnospiraceae (9.98%), and other species in 108 genera (Additional file 3). The number of OUTs unique to the North China zokor was 251, and it was 114 for *M. aspalax* (Figure 3A). The gut bacterial communities of the two species of zokors showed highly significant differences in the abundance-based coverage estimators (ACE) index, Shannon index and Fischer phylogenetic diversity index, and the diversity of gut bacteria of *M. psilurus* was significantly higher than that of *M. aspalax* (Figure 3B–D). The analysis of community composition at the genus level showed that the top 10 genera in the gut bacterial communities fluctuated between species and individuals. The two zokor species’ gut bacterial communities were composed of the three phyla of Firmicutes, Bacteroidota, and Desulfobacterota (Figure 3E). The gut bacterial community of *M. psilurus* mainly contained species from 10 genera, including *Akkermansia* (82.32%) and *Rikenella* (9.05%), and that of the *M. psilurus* mainly contained species from the genera norank_f_norank_o_Izemoplasmatales (62.89%), Bacteroides_ pectinophilus_group (9.10%), and other species from 16 genera (Appendix A).

### 3.3. Analysis of the Association between Food Types and Gut Microorganisms

All plant species were classified into eight types according to the type of plant root system. Mantel analysis of the gut bacterial communities of the two species of zokors with food root types revealed a significant positive correlation between the gut bacterial communities of *M. aspalax* and creeping-rooted plants (Figure 4). The correlation between the gut bacterial community and food type (by root type) of the two species of zokors showed the same trend in terms of statistical significance, although the original hypothesis could not be rejected (Figure 4). That is, the gut bacterial communities of both species of zokors showed a positive correlation with creeping-rooted, bulb-type and axis root-type foods, and a negative correlation with tuberous root-type, sparse clump-type, rhizome-type, fibril root-type and dense clump-type foods (Figure 4).

### 3.4. Indicator Genera and Functional Prediction of Two Zokor Species’ Gut Bacteria

Based on the results of the indicator species analysis, the gut flora of both species of zokors showed the presence of indicator genera. There were 19 indicator genera in the guts of *M. psilurus* and 8 indicator genera in *M. aspalax*. Among them, *Cellulosilyticum*, *Tuzzerella*, NK4A214_group, and *Monoglobus* were the main indicator genera in the gut of the *M. psilurus*, and *Blautia* was the main indicator genera in the gut flora of *M. aspala* (Figure 5).

Based on the results of the test for differences in KEGG functions in the OUT set of the indicated species, the abundances of KEGG functions in the tertiary pathways related to metabolism (primary pathway) were all higher in *M. psilurus* than in *M. aspalax*. The two species of zokors showed highly significant differences in 19 pathways (Figure 6). The functions enriched were related to amino acid metabolism, carbohydrate metabolism and glycogen biosynthesis and metabolism (Figure 6). Eight of these were related to amino acid metabolism, namely, (1) cysteine and methionine metabolism, (2) alanine, aspartate and glutamate metabolism, (3) glycine, serine and threonine metabolism, (4) phenylalanine, tyrosine and tryptophan biosynthesis, (5) valine, leucine and isoleucine biosynthesis, (6) histidine metabolism, (7) arginine and proline metabolism, and (8) lysine degradation (Figure 6). There were 10 pathways associated with carbohydrate metabolism, including (1) amino sugar and nucleotide sugar metabolism, (2) pyruvate metabolism, (3) glycolysis/gluconeogenesis, (4) starch and sucrose metabolism, (5) pentose phosphate pathway, (6) butanoate metabolism, (7) glyoxylate and dicarboxylate metabolism, (8) propanoate metabolism, (9) fructose and mannose metabolism, and (10) galactose metabolism (Figure 6). Only one pathway, peptidoglycan biosynthesis, was associated with glycan biosynthesis and metabolism (Figure 6).

### 3.5. Gut Bacteria Associated with Cellulose Degradation

Because of the herbivory of the two species of zokors, we paid special attention to genes related to carbohydrate metabolism. In rodents, the most abundant gene associated with fiber metabolism was predicted to be beta-glucosidase (K05349). This happened to be present in both zokor gut bacterial communities and differed significantly between the two (Figure 7). In addition to this, two genes, endoglucanase (K01179) and oligosaccharide reducing-end xylanase (K15531), were also significantly different between the two species of zokors, and the abundance of all three genes was greater in the gut bacterial community of *M. psilurus* than that of *M. aspalax* (Figure 7). These results suggest that *M. aspalax* possesses a stronger degradation ability for the cellulose degradation pathway.

### 3.6. Neutral Community Modelling (NCM) of Gut Flora

The neutral community model successfully estimated most of the relationships between the frequency of occurrence of OTUs and their corresponding changes in abundance. The goodness-of-fit values for the gut microbial communities of *M. psilurus* and *M. aspalax* were 66.62% (Figure 8A) and 51% (Figure 8B), respectively. The explanation rate of stochastic processes in the gut flora of *M. psilurus* was higher than that of *M. psilurus*, i.e., stochastic processes were more important in the gut bacteria of *M. psilurus*. The migration rate (Nm) shows that the microbial community spreads in the gut of *M. psilurus* (Nm = 5043) much more widely than in that of *M. aspalax*. (Nm = 2423) (Figure 8A,B). These results suggest that the gut flora of *M. psilurus* has a wider ecological niche than that of *M. aspalax*.

## 4. Discussion

In this study, we compared the gut microbiota and natural foraging preferences of two species of the genus *Myospalax*, distributed in China and living in the Eurasian steppe zone. We focused on the differences in the structure and function of gut bacterial communities between the two species, and analyzed the relationship between microbiota and food types. The results of the study show that the α-diversity of the bacterial community of *M. psilurus* was significantly higher than those of *M. aspalax*, while the species richness, diversity and genealogical diversity of the gut bacterial community in *M. psilurus* were higher than those in *M. aspalax*. In general, diet-associated microbes have a wider range of sources, and along with a more diverse diet, hosts may be exposed to and carry a greater diversity of microbes [65]. On the other hand, the composition of the microbiota is also dependent on the nutrients available in the gut, so a varied diet may increase the α-diversity of the gut microbiota by providing a more diverse range of nutrients [66]. From this we can infer that a richer food composition led to the higher diversity of the gut bacterial community in the North China zokor. In addition, a large number of studies have shown that seasonal factors can have an impact on the food choices of animals and the composition of gut microorganisms [67,68,69]. In particular, for herbivore animals in temperate regions, their access to food resources as primary consumers is severely limited by the biomass of the producers (plants). As seasonal fluctuations in food resources alter their food choices, gut microbial communities also show seasonal dynamics [70]. In autumn, nutrients from perennial plants flow from above-ground parts to roots [71], and the high quality and frequency of foraging help zokors survive the long winter [72]. Therefore, the food diversity in the results of this study may have increased because of the large amount of foraging before overwintering, thus making the microbial diversity higher than that in other seasons. The increase in microbial diversity in the intestinal tract contributes to the more efficient digestion of nutrients.

We identified the genera found among the gut flora of each rodent species. The gut bacterial community of *M. psilurus* mainly contained species from 10 genera, including *Akkermansia* (82.32%) and *Rikenella* (9.05%), and that of the steppe zokor mainly contained species from 16 genera, including norank_f_norank_o_Izemoplasmatales (62.89%), Bacteroides_pectinophilus_group (9.10%) and other species in 16 genera. In studies of the human gut microbiome, bacteria of the genus *Akkermansia* have been found to be involved in mucin degradation [73], and are a probiotic associated with a reduced risk of obesity-related metabolic syndrome [74]. *Rikenella* was found to be the top taxon significantly and positively associated with BMI in a study based on patterns of intestinal flora and body weight changes in wild house mice (*Mus musculus*) at different latitudes [75]. Norank_f_norank_o_o_Izemoplasmatales are considered to be DNA degraders [76]. The abundance of the Bacteroides_pectinophilus_group in enterobacteria was found to be significantly negatively correlated with the prevalence of non-alcoholic fatty liver disease (NAFLD) in a multi-ethnic cohort obesity phenotype study [77]. In conclusion, there were differences in the composition of the intestinal bacterial community between the two species of zokors, and the dominant genera were mainly probiotic-associated groups, with no significant presence of microorganisms that posed a threat to the health of the hosts.

Two kinds of zokor had different indicator genera of intestinal bacteria. *Cellulosilyticum*, *Tuzzerella*, NK4A214 and *Monoglobus* were the main markers in the intestinal tract of *M. psilurus*, and *Blautia* was the main indicator genus in the intestinal bacterial community of *M. aspalax*. It has been shown that *Cellulosilyticum* is associated with fibre and protein breakdown [78]. In preventive medicine studies with naringin regulating the microbiota and metabolome in mice, alterations in *Tuzzerella* were found to be most associated with host endogenous metabolites [79]. NK4A214 of the Ruminococcus family is one of the more abundant genera in the rabbit gut [80], capable of degrading plant polysaccharides to produce volatile fatty acids such as butyric acid [81,82,83], which in turn promotes apoptosis in colon cancer cells and reduces intestinal inflammation [84]. *Monoglobus* is considered to be a highly specialized group of pectin-degrading sugar biota in the human gut [85]. NK4A136 and *Blutia* are beneficial bacteria that produce short-chain fatty acids [86]. *Blutia* is a new genus of Lachnospiraceae that produces short-chain fatty acids (SCFAs) through glucose metabolism and digests dietary cellulose [87], and it is widely found in mammalian feces and intestines, with potential probiotic properties [88]. The functions of the two zokor cecum bacterial community indicator species are involved in plant polysaccharide degradation and the production of short-chain fatty acids, which is in line with the findings of most studies on the function of the gut flora of herbivores [89,90,91].

An analysis of the food habits of the two species of zokors showed that *M. psilurus* mainly fed on *S. officinalis* (68%), while *M. aspalax* preferred a combination of *A. tuberosum* and *Phlomis* sp. (52.1%). In relation to the specialized root-feeding habits of zokors and the root morphology of these three plant species, it is clear that zokors preferred larger food items such as axis root type, bulb type and tuberous root plants, and given that their foraging behavior occurs without the aid of visual searching, this is consistent with what is predicted in the theory of optimal foraging in animals [92]. Environmental heterogeneity shapes the heterogeneity of the distribution of resources available to foragers, and the community structure of natural grassland vegetation clearly follows the same heterogeneity. Optimal foraging theory predicts that an animal’s ability to utilize patches of resources is key to foraging success [93]. As they live underground, zokors are inherently limited in their choice of food resources. Unlike the above-ground parts of plants, root systems are wrapped in a dense medium such as soil, and it is difficult for zokors to weigh the choice of such food resources from multiple perspectives (such as form and color), so a “large amount and easy to obtain” becomes the primary criterion for consideration. Therefore, in their foraging behavior, zokors do not primarily target the dominant species or the established species of the corresponding grassland type to eat, but rather choose species with large root systems as their main food source.

It is well known that herbivores rely on microorganisms living in their gastrointestinal tract to efficiently digest their fiber-rich diets [94,95]. The plant cell wall polysaccharides that make up the bulk of this fiber represent a potentially rich source of carbon [49] that is highly resistant to enzymatic breakdown [96], but certain microbial taxa have evolved mechanisms for degrading sugars from these structural polysaccharides in order to, through intestinal fermentation, gain more access to chemical energy from the diet [97,98]. Short-chain fatty acids (SCFA) are the end products of polysaccharide fermentation produced by gut flora [99,100]. Over a long evolutionary period, animals have optimized their digestive physiology by expanding the volume of either the foregut or hindgut [101,102]. While the microbial communities involved in fiber digestion in foregut fermentation have been well described by studies on ruminant animals [103,104], studies targeting the catabolic utilization of plant polysaccharides have also focused more on foregut fermentation [105,106] or, to a lesser extent, economically viable monogastrics [107,108]. Little is known about the microbial taxa that perform this function in wild herbivores with hindgut fermentation.

Plant cell walls are mainly composed of cellulose, hemicellulose, pectin, and lignin [109,110]. Cellulose is the most abundant of these plant cell wall polysaccharides [111], and its crystal structure makes it one of the most difficult substances to hydrolyze [112]. Therefore, microorganisms in the gut that can efficiently break down cellulose are critical for herbivores’ digestion. We know that host carbohydrate-active enzymes (CAZymes) are mainly produced in the cecum [113]. In the rumen, cellulose degradation is facilitated in part by the efforts of bacterial communities [114], including members of the genera *Fibrobacter* [90] and *Ruminococcus* [115]. Due to the phytophagous nature of both zokors, we paid particular attention to pathways related to carbohydrate metabolism in their gut bacterial communities. Glycosyl hydrolases (EC 3.2.1-) are genes mainly involved in cellulose and hemicellulose degradation [115]. We searched and screened the KO pathways in the OTU set consisting of the two zokor gut bacterial indicator genera based on the “EC 3.2.1-” search condition, and found three pathways (K05349, K01179, and K15531) that were significantly different between the two zokor species. K05349 and K01179 are the most frequently mentioned pathways in relation to the degradation of complex fibers [82]. K05349 is associated with β-glucosidase metabolism, K01179 is associated with endoglucanase metabolism, and K15531 is associated with the metabolism of oligosaccharide-reduced end xylanase. β-glucosidase genes are present in almost all bacterial phyla [116,117].

All organisms involved in cellulose degradation have a cellulase system consisting of a multi-enzyme complex of three enzymes: exoglucanase (also known as cellulobiohydrolases, CBH, EC 3.2.1.91), endoglucanase (EC 3.2.1.4), and β-glucosidase (BGL, EC 3.2.1.21), which work synergistically for the complete hydrolysis of cellulose [118]. Therefore, it is not surprising that K05349 and K01179 appeared in the metabolic pathway of the two zokor species’ gut bacteria, but CBH2 (K019668), a gene related to cellulobiohydrolases, showed very low abundance, and CBH1 (K01225) was absent in the gut bacterial community of zokors. This may be due to the fact that cellulases are widely present in natural organisms and are produced by bacteria, fungi and protozoa [119,120]. However, the cellulases produced by bacteria are mainly glucan endonucleases, so it is reasonable that genes related to cellulobiohydrolases are rarely expressed in the bacterial communities of the zokor gut. In a study of five desert rodent gut bacterial communities, predictions of the cellulose degradation function showed that K05349 and K01179 were the two most abundant genes in these rodents’ gut bacterial communities [81]. The KEGG pathways predicted from the gut flora of those five different dietary species of desert rodents and this study’s two specialized subterranean rodents suggest that the cellulose degradation pathways of K05349 and K01179 act as “generalists” in the rodent gut bacterial community. β-Glucosidase is an important hydrolase that catalyzes the hydrolysis of β-glucosidic bonds to release glucose molecules [121]. In the gut of monogastric animals, this enzyme helps to break down dietary carbohydrates, facilitating energy acquisition and utilization, and is able to specifically catalyze the hydrolysis of cellulose [122]. Endoglucanase breaks down cellulose by cleaving the β-1,4-glucosidic bonds within its long chains into smaller oligosaccharides [123]. Oligosaccharides are substances that can be utilized by probiotics to promote a balanced intestinal flora, thereby enhancing host health [124]. In addition to the two pathways mentioned above, the metabolic pathway associated with K15531 also differed between the two species of *Myospalax*, but its abundance was much lower than that of the two pathways mentioned above. Oligosaccharide-reduced end xylanase (K15531; EC3.2.1.156) is a high-molecular-mass xylanase that degrades xylan, a dietary fiber in plant cell walls [125]. Xylan is the most common hemicellulose component of grass, leaves, straw, and wood, the second most abundant renewable resource on earth [126,127], a major component of hemicellulose [128], and a polysaccharide component of up to 45% of ruminant feed [129]. Of interest, this pathway has not been reported in intestinal bacterial studies in herbivorous mono-gastrics, but rather in a study based on obesity complications in US immigrants, which concluded that levels of oligosaccharide-reducing end xylanase (K15531) increased with fiber intake, which was negatively correlated with the severity of obesity [130]. These results imply that the process of cellulose degradation by the bacterial community in the gut of zokor includes not only pathways that are prevalent in mammals, but also unique degradation pathways. Among these cellulose degradation pathways, *M. psilurus* possessed a stronger degradation capacity.

Based on the estimation of the neutral community model of the intestinal bacterial communities of the two species of zokors, it was found that the importance of stochastic processes in the gut bacteria of *M. psilurus* exceeded that of the *M. aspalax*, and that the mobility of the intestinal bacterial community of *M. psilurus* was greater than that in *M. aspalax*, i.e., the microbial community spread further in the intestines of *M. psilurus* than in those of *M. aspalax*. Combined with the differences in other ecological characteristics (including the number of OTUs, α-diversity, and KEGG function prediction) of the gut bacterial community of *M. psilurus* between the two, as described above, we can conclude that the intestinal bacterial community of *M. psilurus* has a wider ecological niche compared with that of *M. aspalax*. The results of this paper clarify the diet and gut microecology of two subterranean rodents, but do not incorporate the effects of season on food composition and gut microbial communities in zokors. Given the ability of season to shape diet and gut microbiology, we expect to further investigate the response of gut microbials to seasonal variations in subsequent studies through monitoring on longer time scales, and to identify and incorporate other influences on diet and microbiology.

## 5. Conclusions

In this study, we investigated the food composition and gut microbiological characteristics of two species of zokors. Gut microbial bacterial community diversity was significantly higher in *M. psilurus* than in *M. aspalax*, and the two zokor species possessed different gut bacterial indicator genera. *Cellulosilyticum*, *Tuzzerella*, NK4A214_group, and *Monoglobus* were the main indicator genera in the gut tract of *M. psilurus*, and *Blautia* was the main indicator genera in the gut flora of *M. aspalax*. Based on the classification of root types of food source plants, there was a significant positive correlation between the gut bacterial communities of *M. aspalax* and the rhizome-type plants, and there was a complementary relationship between the axis root-type and rhizome-type-plant-dominated (including bulb type and tuberous root food) food groups. In the cellulose degradation pathway, *M. psilurus* possessed a stronger degradation ability. Stochastic processes play a more important role in the gut bacterial community of *M. psilurus*, suggesting a wider microbial niche for *M. psilurus*. The characterization of the diet and gut microecology of two species of zokors provides a new perspective on the relationship between diet and gut symbiotic microorganisms in subterranean rodents in the steppe zone.

## Figures and Tables

**Figure 1 animals-14-02334-f001:**
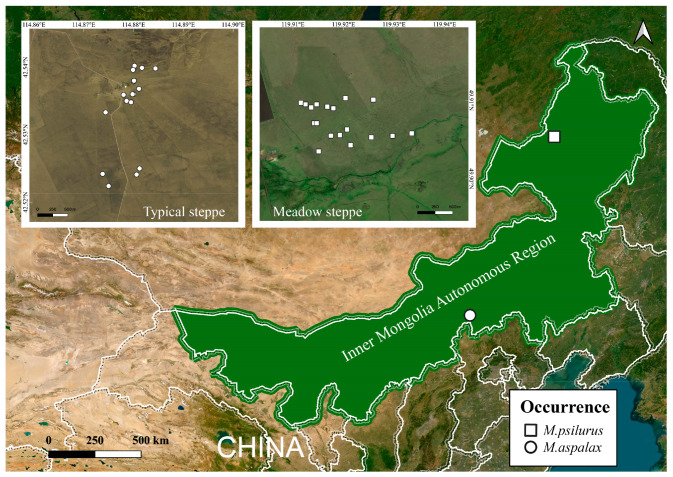
Schematic diagram of the sampling points for two species of zokors. “□” indicates the capture location of *M. psilurus* and “○” indicates the capture location of the *M. aspalax*.

**Figure 2 animals-14-02334-f002:**
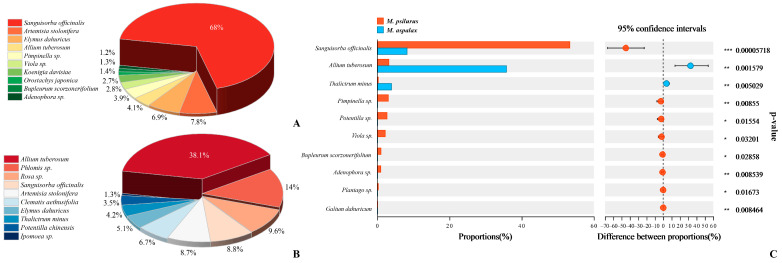
Food composition of the two species of zokors. (**A**) Top 10 species in the diet of *M. psilurus*. (**B**) Top 10 species in the diet of *M. aspalax*. (**C**). *t*-test for between-group differences in food composition between the two species of zokors, with *p*-values corrected using the Fdr method, and confidence intervals calculated using Welch’s t-method, with a confidence level of 0.95. *p*-values after correction are shown on the right side, denoted as * 0.01 < *p* ≤ 0.05, ** 0.001 < *p* ≤ 0.01 and *** *p* ≤ 0.001.

**Figure 3 animals-14-02334-f003:**
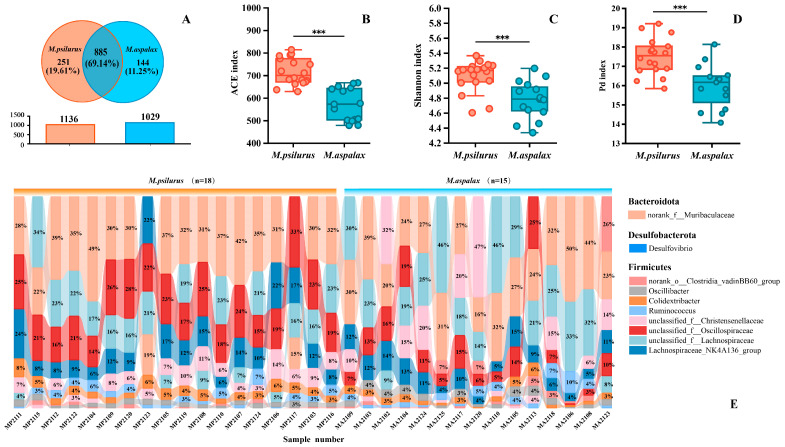
Microbial diversity and species composition in the guts of two species of zokors. (**A**) Venn diagram showing the percentages of OTUs common and unique to the two species of zokors. (**B**) Test of intergroup differences in ACE index of the gut bacterial community of the two species of zokors. (**C**) Test of intergroup differences in the Shannon index of the intestinal bacterial communities of the two species of zokors. (**D**) Test of intergroup differences in the Pd index of the intestinal bacterial communities of the two species of zokors. (**E**) Composition of the two zokor species’ intestinal bacterial communities at the genus level. Significance levels are denoted as *** *p* ≤ 0.001.

**Figure 4 animals-14-02334-f004:**
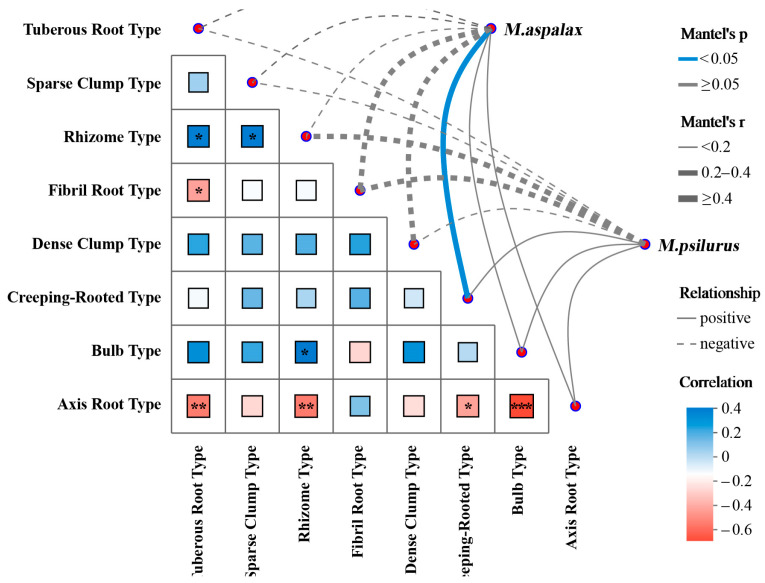
Mantel analyses of food types and gut bacterial communities of the two species of *Myospalax*. The color of the line indicates the *p*-value of the correlation between the gut microbial community and the food composition distance matrix (blue: *p* < 0.05; gray: *p* ≥ 0.05). The thickness of the line indicates the strength of the correlation between the microbial community β-diversity distance matrix and the food composition distance matrix (wide: r ≥ 0.4; medium: 0.4 ≥ r ≥ 0.2; thin: r < 0.2). The solid lines indicate positive correlations and dashed lines indicate negative correlations. The colors of the boxes reflect the strength and direction of correlation between food types, with blue representing a positive correlation and red representing a negative correlation. * 0.01 < *p* ≤ 0.05, ** 0.001 < *p* ≤ 0.01 and *** *p* ≤ 0.001.

**Figure 5 animals-14-02334-f005:**
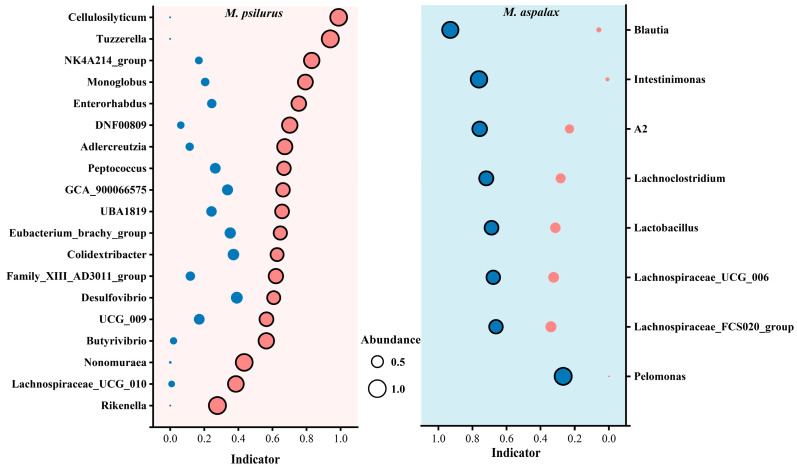
Abundance and indicative values of bacterial indicator genera in the gut flora of two species of *Myospalax*. Indicator genera are listed on the *y*-axis, and their indicator values are represented on the *x*-axis. The colors of the circles and the background represent the two species of zokors, pink for *M. psilurus* and blue for *M. aspalax*, with the size of the “○” indicating the abundance of the indicator genera.

**Figure 6 animals-14-02334-f006:**
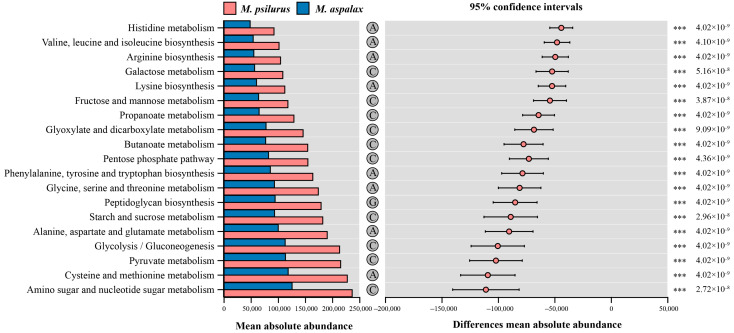
Functions of metabolism-related intestinal bacterial communities, including differences between the two species of *Myospalax*. The *x*-axis of the left bar graph indicates the average absolute abundance of the three-level pathways in different subgroups, and the *y*-axis lists the names of the pathways. The circles with letters on the right side of the bar graph represent the secondary pathway to which the corresponding functional pathway belongs: A for amino acid metabolism, C for carbohydrate metabolism, and G for glycan biosynthesis and metabolism. The middle area is the confidence interval, and the dots represent the average absolute abundance of the species in the two groups. The bars on the dots are the upper and lower limits of the confidence intervals for the difference, and the right-hand side is the corrected *p*-value, *** *p* ≤ 0.001.

**Figure 7 animals-14-02334-f007:**
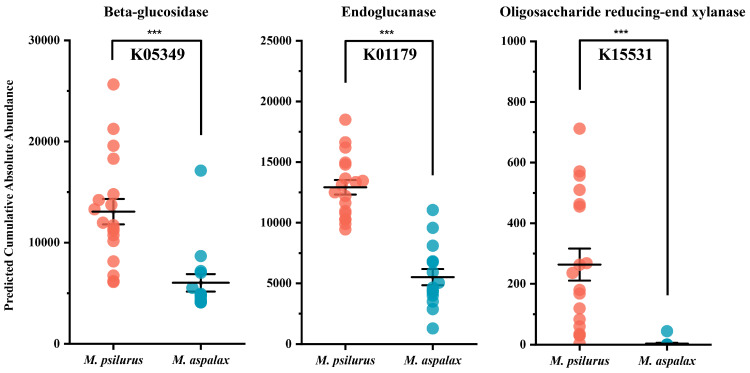
Predicted abundance of cellulose degradation-related genes. The *y*-axis represents the predicted values of absolute abundance of related genes. *** *p* ≤ 0.001.

**Figure 8 animals-14-02334-f008:**
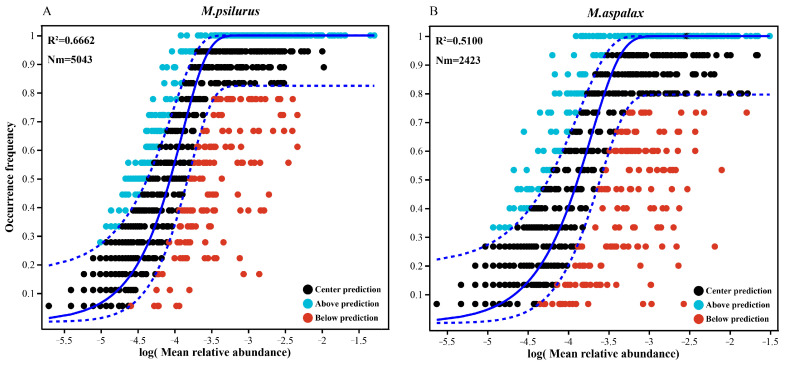
Estimation of the neutral community model for the gut bacterial communities of the two species of zokors. The *x*-axis represents the logarithm of the average relative abundance of species, and the *y*-axis depicts the frequency of occurrence. The solid line represents the fit of the neutral model, and the upper and lower dashed lines represent the 95% confidence of the model prediction. R^2^ represents the overall goodness-of-fit of the neutral community model, and the higher R^2^ indicates the model is closer to the neutral model, which means that the construction of the community is more affected by stochastic processes, and less by deterministic processes. Nm is the product of metacommunity size (N) and migration rate (m) (Nm = N × m), which is used to assess the degree of dispersal among communities.

## Data Availability

The datasets presented in this study can be found in online repositories. The names of the repository and accession number(s) can be found below: Sequence Read Archive (NCBI, USA), PRJNA1089538.

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
