# Peer review of "Natural Foraging Selection and Gut Microecology of Two Subterranean Rodents from the Eurasian Steppe in China"

_animals, 2024, doi:10.3390/ani14162334_

Round 1

Reviewer 1 Report

Comments and Suggestions for Authors

1, The section 1. Introduction is too long. To be more logical and layered, this section 1. Introduction can be appropriately divided into smaller paragraphs.

2, The research objectives need to be further refined and rewritten.

3, What is the author's reason for using the cecum contents to study the these gut microbiome?

4, Some M. aspalax in the section 3. Results should italic.

5, The corresponding calculation method of Neutral Community Modelling in the section 3. 6 should be supplemented into the section  2. Materials and Methods”.

6, The diet  of animals often include many contents, such as feeding objects, food nutrient composition, feeding range and feeding pattern, etc. So, The authors should define diet in the introduction or somewhere else.

Reviewer 2 Report

Comments and Suggestions for Authors

The authors provide an insightful investigation into the gut microecology of two subterranean rodents, Myospalax psilurus and Myospalax aspalax, from the Eurasian steppes in China. The study uses 16S amplicon and macro-barcoding technologies to analyze the gut bacterial communities and diet composition of these rodents. The authors found significant differences in gut microbial diversity and composition between the two species, highlighting the relationship between their diet and gut microbes. I have a few comments.

Major Comments

1. The paper mentions functional predictions based on KEGG pathways, indicating stronger cellulose degradation abilities in M. psilurusHow do these functional predictions translate to actual physiological differences in the two rodent species? Are there observable phenotypic differences that support the microbial functional predictions?

2. The study uses 33 individual rodents, with a slightly imbalanced representation of the two species. How did you ensure that this sample size is sufficient for robust conclusions?

3. Were there any seasonal variations in diet or gut microbiota composition considered during the sampling period? Given that the samples were collected in autumn, it would be important to discuss potential seasonal effects on the results and how they might influence the conclusions.

Minor Comment

Figures and tables are well-presented, but some figures' resolution can be improved. Like Figure 2 and Figure 3.

Thank you!

Reviewer 3 Report

Comments and Suggestions for Authors

1) Most microorganisms residing in the gut have 51 profound effects on host physiology and nutrition, which are critical for host health. Cite some more references.

2)  Food habits reflect the adaptability 65 and ecological role of animals. Complete the sentence.

3) Mention exclusion and inclusion criteria in materials and methods.

4) Each sample was replicated three times, and the PCR products from the same sample were mixed and detected 122 by 2% agarose gel electrophoresis. It is 5% agarose gel. 

5) The number of mismatches allowed by the barcode 143 was 0, and the maximum number of primers mismatched was 2. Correct the sentence.

6) α-diversity. Describe it in the introduction.

7)  The analysis of 225 community composition at the genus level showed that the top 10 genera in the gut bacterial communities were significantly different between the zokor species and also varied 227 by individual within a species. Make the sentence concise. 

8) Improve the conclusion section. 

Comments on the Quality of English Language

There are some linguistic and Grammatical corrections required. 

Reviewer 4 Report

Comments and Suggestions for Authors

The paper by Shang et al. presents experimental data on bacterial communities of 2 rodents in unique sites based on metabarcoding procedures. The paper is well written, organized and adds new understanding on gut microecology. Good experiment design, all methods used are detailed and fully spelled out. However, there are comments regarding the interpretation of the data. The manuscript gives ample attention to the cellulose degradation by the bacterial communities studied. However, there is no information on the genes for important enzymes such as cellulobiohydrolases (CBH, EC 3.2.1.21), they cleave crystalline regions in cellulose. The main product of their catalytic reactions is cellobiose, which as a result of synergistic action is destroyed to glucose by b-glucosidases, which are present in the studied system on the basis of the KEGG method. Describing the degradation of another polymer, xylan - only oligosaccharide reduced-end xylanase is found, but there is no discussion of the importance of endoxylanases. Since they are essential for the production of oligosaccharides. At least some comments is needed on these two observations.

In addition, lines 453-464 (about xylanases) require some reduction and deletion of repeats. It is possible to specify and indicate the amount of this polymer in the studied rodent foodstuffs. The quality of Fig. 2 also needs to be improved.
